# A Method for the Study of Cerebellar Cognitive Function—Re-Cognition and Validation of Error-Related Potentials

**DOI:** 10.3390/brainsci12091173

**Published:** 2022-09-01

**Authors:** Bo Mu, Chang Niu, Jingping Shi, Rumei Li, Chao Yu, Kuiying Yin

**Affiliations:** 1Nanjing Research Institute of Electronic Technology, Nanjing 210019, China; 2Neurology Department of the Affiliated Brain Hospital of Nanjing Medical University, Nanjing 210029, China

**Keywords:** electroencephalography (EEG), error-related potentials (ErrP), cerebellar regions, classification method

## Abstract

The cerebellar region has four times as many brain cells as the brain, but whether the cerebellum functions in cognition, and how it does so, remain unexplored. In order to verify whether the cerebellum is involved in cognition, we chose to investigate whether the cerebellum is involved in the process of error judgment. We designed an experiment in which we could activate the subject’s error-related potentials (ErrP). We recruited 26 subjects and asked them to wear EEG caps with cerebellar regions designed by us to participate in the experiment so that we could record their EEG activity throughout the experiment. We successfully mitigated the majority of noise interference after a series of pre-processing of the data collected from each subject. Our analysis of the preprocessed data revealed that our experiment successfully activated ErrP, and that the EEG signals, including the cerebellum, were significantly different when subjects made errors compared to when they made correct judgments. We designed a feature extraction method that requires selecting channels with large differences under different classifications, firstly by extracting the time-frequency features of these channels, and then screening these features with sequence backward feature (SBS) selection. We use the extracted features as the input and different event types in EEG data as the labels for multiple classifiers to classify the data in the executive and feedback segments, where the average accuracy for two-class classification of executive segments can reach 80.5%. The major contribution of our study is the discovery of the presence of ErrP in cerebellar regions and the extraction of an effective feature extraction method for EEG data.

## 1. Introduction

The cerebellum is located at the back and lower part of the cerebrum, and is comprised of the cortex, cerebellar nuclei, and conduction links between the cerebellum and other regions. The cerebellum accounts for about 10% of the volume of the whole brain, but it has more than 70 billion neurons, which is about four times the number of brain cells in the brain, and males have slightly more than females [1,2,3]. The cerebellum plays an essential role in sensory perception and coordination; it is also involved in many cognitive functions such as attention and language, and can also regulate responses such as fear and joy. The best known of these are the functions related to movement. The cerebellum is responsible for processing motor patterns and coordinating muscle movements, maintaining body balance, processing cognitive patterns, and automating certain repetitive tasks such as breathing and heartbeat [4]. The cerebellum is also involved in cognitive activity at some level. Evidence for this is provided by the fact that, during the evolution of the human brain, some new parts of the phylogeny evolved and expanded in the cerebellum, in parallel with the expansion of the association areas in the cerebral cortex. Moreover, the development of these two structures is a prerequisite for the advanced information processing capabilities that distinguish humans from other primates or animals [5]. Another strong piece of evidence comes from anatomy. In studies of people with autism, the most consistent pathological site was the cerebellum after magnetic resonance imaging (MRI) or computed tomography (CT) scans. In most studies, patients were found to have abnormalities only in the cerebellar region [6]. However, the specific effect of the cerebellum in cognition has not been conclusively established.

Several studies have shown that subjects elicit a characteristic EEG event-related potential (ERP) after making a mistake in a response choice task or when they observe feedback results that do not meet expectations, among several other situations. This kind of ERP is called error-related potential (ErrP) [7,8]. ErrP is usually found in the anterior cingulate cortex, a brain region involved in emotional and attentional processing, so it is usually found in the central and prefrontal locations of the EEG [9]. The ErrP has recently been utilized as an ERP component that can be used to correct BCI errors [10,11]. In the choice-response task, subjects responded to stimuli by pressing a button, and error button presses produced “execution error”. ErrP that occurs when a user performs an error in a reinforcement learning task and receives feedback indicating the error action is often referred to as a “feedback error” [12]. Still, the form of latency and potential varies with experimental paradigms [13].

Go/No go is a standard experimental paradigm in psychology. It is a widely recognized and stable cognitive experiment. It requires the subject to respond between performing or inhibiting the motor response. The stopping task requires the inhibition of a motor response triggered by a “stop” signal shortly after the start signal, thus converting the start response posterior to a forbidden response, involving the withdrawal of a response already triggered by the Go signal [14]. Other experiments have shown that various regions of the brain are related to behavioral execution and behavioral inhibition in the Go/No go task, including the orbital gyrus, inferior frontal gyrus, medial frontal lobe, temporal lobe, parietal cortex [15,16,17,18]. Based on the current studies, we hypothesized that the cerebellum is involved in cognitive processes, and we verified this by using methods that detect the presence of ErrP in cerebellar regions.

Commonly used research methods of cerebellar regions, such as fMRI and CT, have disadvantages such as poor immediacy and difficulty in performing tasks simultaneously. Thus, we wondered if we could design an EEG cap that could cover cerebellar regions to test the presence of ErrP in cerebellar regions and to demonstrate whether cerebellar regions are involved in cognitive functions. We designed, outsourced for production, and used the EEG cap as shown in Figure 1, including electrodes for cerebellar regions such as CBz, CB1, and CB2 [19]. We designed a Go/No Go experiment and recorded the subjects’ EEG throughout to verify whether the experimental paradigm could induce ErrP, whether ErrP was present in cerebellar regions and whether these obtained data could be well classified for further BCI design. Specifically, we recruited 26 subjects. The subjects’ EEG data were tracked throughout the experiment using an EEG cap of our design. Each person performed 500 trials of easily induced ErrP as in Figure 2. Afterwards, we pre-processed, analyzed, and classified their data to investigate whether there were ErrP in the cerebellar regions and whether error or correct conditions could be effectively classified based on the data obtained.

## 2. Methods

### 2.1. Task Description

The experiment consists of image judgment and Go/No go mechanism, which enables the subjects to make more mistakes and be induced enough execution errors and feedback errors. The experimental process of each trial is shown in Figure 3. Each trial of experiments includes the following four steps: (1) A black “+” appeared in the center of the screen for 800 ms, prompting the subjects to concentrate and recover from the last judgment; (2) The stimulus image displayed in the center of the screen for 1000 ms; (3) After the stimulus picture disappeared, the black “+” was illuminated for another 800 ms, during which the subject needed to make a decision whether or not to press the key, which we call the 800 ms response window; (4) Different feedback images appeared based on the subjects’ performance in this trial. In the second step, the chances of appearing images with target are three times higher than without target. During the third stage, the subject was instructed to press the space bar when they detected an existing target on the stimulus that was just presented. In approximately 20–80% of the trials with targets, a “STOP” sign would appear in about 150–650 ms. When the subject encountered these trials, he or she was instructed not to press the space bar even if there was a target. The subject’s prior overall performance and recent performance determined the frequency and delay of the sign. Simply put, the higher the accuracy, the later and the more the sign appeared. The fewer the number of keystrokes, the lower the chance of the sign appeared. These measures were done to elicit behavioral errors for the purpose of this experiment. Each subject was given 10 opportunities to practice before each experiment. In order to ensure the number of samples in the experiment, each subject conducted 5 rounds of experiments, each round of experiments included 100 trials, and the subjects freely chose a certain time to rest during each round.

In the trial, there were three possibilities of the stimulus picture occurring: contain the target normally, contain the target but have the “STOP” sign, and do not contain the target. These were labelled the target trial, stop trial, and no-target trial, respectively. When collecting the data, we divided the data of the execution segment into the following four classes: (a) Subject pressed the key in the target trial; (b) Subject pressed before the sign appeared in the stop trial; (c) Subject pressed after the sign appeared in the stop trial; (d) Subject pressed in the no-target trial. The data in the feedback segment is divided into the following three classes: (a) overtime, subjects did not press the key before the end of the response window for the target trial; (b) correct, the correct feedback image will appear in the following cases, including (1) subject pressed the key correctly in the target trial, (2) subject did not press in the stop or no-target trial; (c) error, subjects pressed in stop trial or no-target trial. Each stimulus picture was added with a certain degree of Rayleigh noise and gamma noise to increase the difficulty of experimental judgment.

### 2.2. Subjects and Devices

A total of 26 male subjects were included in this study. All of them were informed and agreed to the experimental content before participating. The average age was 26.7 ± 4.2 years old. All subjects were in a healthy condition, and right-handed. There was no color blindness or sensory impairment. Two subjects withdrew from the experiment after 300 experiments. Subjects were engaged with a high level of attention throughout the experiment. The experiment was carried out in the EEG collection room of the Affiliated Brain Hospital of Nanjing Medical University. We used an EEG cap with 69 + 2 channels (Neuracle Technology (Changzhou) Co., Ltd., Changzhou, China) [19]. The electrode positions were placed according to the 10–20 international standard lead system (Figure 4), with 69 fixed electrodes, one reference electrode at the CPz position in the standard, and one ground electrode at the AFz position in the standard. In Figure 4B, the blue, red and green lines respectively represent the nasal tip direction, the cranial vault direction and the left ear direction. The center of the parietal lobe is taken as the origin of the coordinate axis, the positive *x*-axis is the front of the human brain, and the positive *y*-axis is the left of the human brain. All data were sampled at 1000 Hz.

### 2.3. Pre-Processing

The data file collected by each subject contains two parts: data and events. The two files respectively record all waveforms of 69 electrodes, and the time and type of events. These two parts of the data are read by specific extensions of the EEGLAB 14.0 toolbox of MatLab R2020b. Figure 5 shows the data after direct reading, which is the EEG waveform of CBz recorded by subject 03 during the whole process of an electrode experiment. Similar to other subjects and other electrodes, we can see that the EEG signal is more disordered and fluctuates significantly over time. We need to pre-process it to rectify it, remove noise, and segment it into a readable signal.

The first step is to select the channel. It is necessary to exclude the unused SP1 and SP2 electrodes. If there are broken channels in the experiment, they also need to be removed. Then we need to import the location information, as shown in Figure 4, and associate the location with each electrode to locate the brain area conveniently. The third step is re-reference. We use the most common method, the global averaging method. It does not require any other data than the electrical signal and effectively eliminates the effect of the reference electrode position. In addition, re-referencing can attenuate interference and bioelectric effects in the system [20]. Then we filter the signal with a 48–52 Hz band-stop filter to remove the 50 Hz power frequency AC interference. We then use a 0.1–50 Hz band-pass filter. The purpose of high-pass filtering at lower cutoff frequencies is to eliminate baseline drift while preparing for independent component analysis (ICA) [21]. The cut-off frequency for low pass filtering is 50 Hz. The first reason is the common EEG band at the range of 0.5–30 Hz (Delta wave: 0.4–4 Hz; Theta wave: 4–7 Hz; Alpha wave: 8–12 Hz; Beta waves: 13–30 Hz) [22,23,24]. Second, because the main energy of the surface electromyography (sEMG) was concentrated in the band 20~150 Hz, a certain low-pass filter can effectively filter out the influence of sEMG.

After the previous processing, we start intercepting the data, and in Section 2.1, we introduce the classification method of execution and feedback. We carved out 500 ms before and 1000 ms after the event as the time segment of interest, based on key press time and feedback image appearance time. Each subject can generate 500 feedback waveforms and several execution waveforms. Then we reduce the frequency of the data from 1000 Hz to 250 Hz and execute ICA. ICA can effectively remove artifacts such as eye movements. The rows of the input matrix X for independent component analysis are the EEG signals of different electrodes, the columns are the measured values recorded at different time nodes, and ICA finds a decomposition matrix W after many iterations, forming the equation: U = WX, converting X on the time scale to Y on the component scale. The new matrix W imparts different scalp topography to each component and provides evidence for the physiological origin of these components [24]. Figure 6 shows the topographic map of each component after decomposition. For example, component 1 is an obvious eye movement artifact that needs to be removed. We took the average value of the first 500 ms of each segment as the baseline value and subtracted this baseline data from the entire segment to keep the data signal of each segment around the 0 potential. The execution waveform data of the electrode shown in Figure 5 after processing is shown in Figure 7. The three sets of waveforms respectively represent the waveforms of the three cases of correct (a), pending (b), and error (c, d) in execution. All data pre-processing steps for each individual subject are shown above [24].

### 2.4. Method of Classification

We use a method that selects specific channels, extracts specific time-frequency domain features, and then combines the two to perform feature selection using the sequential backward selection (SBS) method to select the optimal number of features and feature types, and uses support vector machines (SVM) and K-Nearest Neighbor (KNN) methods to target the final feature matrix for classification.

SVM classifier: The SVM technique is a computer algorithm based on statistical theory to learn the labels assigned to objects. These support vectors try to find a hyperplane that separates the different classes of points in the hyperspace [25]. The SVM classifier was implemented in python using scikit-learn packages. We set the hyperparameter C, which controls the error of the class separation, to 0.1 and the kernel function parameter to “poly”, which means polynomial.KNN classifier: The KNN classification algorithm is one of the data classification technique. The principle is that if the majority of the K nearest neighboring samples in the feature space belong to a certain class, then the sample also belongs to that class, rather than relying on the discriminative class domain [26]. The SVM classifier was implemented in python using scikit-learn packages. We set the number of hyperparameters K to 3.

For each channel in each trial of each person, we extracted 15 common time-frequency eigenvalues. Including time-domain features: maximum value, maximum value appearance time, minimum value, minimum value appearance time, channel mean value, kurtosis, skewness, standard deviation, area (means the definite integral of the electrode signal during this 1500 ms), peak to peak distance (represents the difference between the time of occurrence of the maximum and minimum values of the electrode), and zero-crossing rate (means the percentage of time electrode signals greater than 0 in this segment). Frequency-domain characteristics: energy in the four frequency bands of Delta, Theta, Alpha, and Beta waves. In the case of research feedback error, For the feedback segment, the 15 features of the 69 electrode channels of these 500 trials form the matrix. Similarly, because the subjects did not execute every trial, we composed a 69 × 15 × N feature matrix (the number of N varied from subject to subject) when studying the execution segments. We cross-correlate the average value of the correct trials with the error trials. We can consider that the channel with a lower correlation is the channel with a larger proportion of components participating in the generation of the ErrP. We combine the selected channel with the previously formed feature matrix, extract the feature matrix of the selected channel, and perform SBS on the basis of the new feature matrix to select the optimal feature subset. Doing so picks out the most influential features and avoids the curse of dimensionality. We use sequence backward selection for further filtering of features, which is a typical greedy sequence feature selection algorithm. SBS sequentially removes features from the full feature subset until new feature subspace contains the desired number of features. At each stage we need to eliminate the features that cause the least performance loss after removal [27]. In our approach, we sequentially remove one of the remaining N features at each stage, and then use the remaining N-1 features as input to the KNN classifier. We can eliminate at each stage the features that have a negative effect or minimal positive effect on the classifier in this way, until we find the remaining features that work best for classification. To ensure that there are no extreme cases, such as the final selection of very few feature values, we restrict the number of features selected to be greater than 10, in other words, the number of features with the highest correct KNN classification rate is selected while the number of remaining features is greater than 10. Then we take the optimal feature matrix as input and use a support vector machine to classify it. The support vector machine is a classification model, which is a linear classifier defined in the feature space such that the interval between different sets is maximized by determining a separation hyperplane such that the samples of different classes are on different sides of the hyperplane and the sum of the distances from the support vectors of different classes to the hyperplane is maximized. For the training set:
T = {(*x*_1_, *y*_1_), (*x*_2_, *y*_2_), …, (*x*_n_, *y*_n_)}, *x*_i_ ∈ X = R^n^, *y*_i_ ∈ {1,2,3}(1)

*x*_i_ and *y*_i_ respectively denote the different dimensions of the feature space and each feature value after normalization, and the separation hyperplane is
*w*^T^*x* + *b* = 0(2)

Assuming that *x*_i_ is the support vector, the functional margin from other sample points to the separation hyperplane is necessarily greater than or equal to the functional margin from the support vector to the separation hyperplane (meaning that the classification result does have confidence). The basic idea of SVM is to solve a separating hyperplane, so that the geometric interval between different sets can be maximized and can be separated correctly. It can be regarded as a constrained optimization problem:(3)$$\min\frac{1}{2}{\left\|w\right\|}^{2},s.t.\;y_{j}\left(w^{T}s_{j}+b\right)-1\geq 0$$

Finally, the feature space is partitioned by the obtained hyperplane, and the classification result of the sample is decided based on the position of the sample located in the partitioned feature space. In addition, the fault tolerance for nonlinear classification and models can be improved by constructing slack variables for the sample points. Different kernel functions can also be introduced to change the type of kernel of the SVM classification function [28,29].

Our flowchart for pre-processing, feature extraction and classification of EEG data is shown in Figure 8.

## 3. Results

First, we optimized a dedicated classifier for each subject. The process of processing and analysis is the same for each subject, but the results are different, so we choose subject 03 as an example to illustrate. We study the waveform data of his execution segment. In 500 trials of experiments, he pressed a total of 194 times, of which 88 were correct, 36 were errors, and 50 were pending. Correctly corresponds to (a) of the behavior section in Section 2.1, error corresponds to (c) and (d), and pending corresponds to (b). We split his training set and test set in a ratio of 7:3, and the ANOVA result for mean values of the three cases in the training set are shown in Figure 9. It can be seen that, in terms of ANOVA, there is some difference between these three classes. More generally, we subtracted the mean of the correct and error cases at the CBz electrodes of the cerebellum of multiple subjects in Figure 10. Here, the error-related activity is colored in blue, and the red waveform represents the averaged EEG activity when a correct response was made. The black plot shows the difference between red and blue (error vs. correct). We can see that there are obvious differences between the two, which indicates that the cerebellum has ErrP and is involved in cognitive activities to a certain extent. 

We use the analysis of the execution segment of subject 03 to illustrate our process of filtering features. We first need to have screened the channels. After the cross-correlation of the average waveforms of the correct and error cases, the correlation of each channel is shown in Figure 11. We set a correlation of 0.7 as the threshold and selected out 13 channels with low correlation as candidates. For subject 03, these channels were C3, C4, F8, T8, CP1, CP2, FC5, C2, CP4, C5, C6, FT7, FT8. The parietal lobe occupies more components, which is in line with the general research results of ErrP. The electrodes on the left and right sides are distributed evenly, which means that the influence of the keystrokes on the selected channels is low. The selected 13 channels are combined with 15 eigenvalues to obtain 195 eigenvalues. We then used the SBS algorithm to select the most appropriate feature matrix. The SBS selection channel is performed on these 195 variables, which are divided into three cases: correct, error, and pending. Each iteration performs multiple three-class KNN and removes an eigenvalue with the most negligible positive impact. The final classification accuracy rate change curve is shown in Figure 12. The curve has the advantage of high correct rate with small channels of KNN classification for the remaining 19 features. The 19 screened features contain different types of time-frequency features for various channels. We use KNN as a classifier for three-class classification task for correct, error and pending. When we use all features of all channels without filtering, the accuracy is 73%. When using the filtered features, the accuracy is 83%. When we use SVM as classifier and filtered features as input, the accuracy is 84%.

We now study only the correct and error cases, excluding the interference pending cases, for KNN classification of two classes. In the same way, we use SBS for those 195 features. We have achieved good results in every number of features in the training set, so 11 features were selected as thresholds. They are time domain features of the top lobe mainly. We performed KNN classification on the full 195 features in the training set and obtained 97% accuracy, and obtained 99% accuracy on the selected 11 features. Similar results were obtained when filtering and classifying others, although the channels and features obtained were not the same.

We ran the method 10 times for multiple verifications and randomly allocated the training set and the test set at a ratio of 7:3 each time. The average accuracy and standard deviation obtained from each trial are shown in Table 1 below. For the test set, the three-class SVM, KNN after feature extraction, and the full-feature three-class SVM, KNN have the accuracy of 37.8%, 33.2%, 34.1% and 31.7%. We can conclude that, since the channels and features are selected in the training set, the accuracy in the test set is almost the same as random, although the correct classification rate in the training set is nice. The classification method and classification model of three-class classification have no utility. The four accuracies of the two-class classification are 82.1%, 84.7%, 34.1%, and 45.2%, respectively. It can be seen that the above method has certain practical value in the two-class classification. The accuracy rate after feature extraction has increased by 36.9 and 40.6 percentage by two methods. As can be seen from the table, our approach of feature filtering has good adaptability for a wide range of classifiers with certain enhancement effects.

Similarly, we analyzed the feedback errors of the same subject 03, with a total of 500 trials, including 332 correct trials, 98 error trials, and 70 overtime trials. The same approach is taken as for the execution segment step. For subject 03, the selected channels were C3, C4, Cz, CP2, FC5, FC6, FC4, F6, C5 and FT7, which are similar but not identical to the execution segment, as expected. The accuracy reaches a maximum at the remaining 11 features, which are mainly the time-domain features of the electrodes in the parietal lobe. When we perform two-class classification after excluding the overtime case, the situation is similar to the execution segment, so we choose the ten features as the threshold. We execute the method several times, using 350 random samples for training and 150 samples for testing. The average accuracy and standard deviation obtained are shown in the following Table 2. Compared with the execution segment, the accuracy of the feedback segment for the two-class classification has decreased, but the KNN and SVM still improved by 8.1 and 21.9 percentage points after screening, which shows the positive effect of the screening method. For the three-class classification, the accuracy improvement of KNN is small and that of SVM is 26.0%, illustrating the generalizability of our method again. We performed the above steps for feature screening on all subjects, and the pre-screening and post-screening accuracy of SVM for each case are shown in Table 3.

As we can see in Table 3, there are some differences in the results for each subject. Before filtering the features, the accuracy of both the execution and feedback segments was around 50% for the two-class classification and 33% for the three-class classification. After the feature screening, for the execution segment, the average accuracy of the two-class classification rises from 50.1% to 80.5%, which shows the remarkable screening effect; the three-class classification goes from 36.0% to 39.6%, with almost no change. We suspect that this is because the third of the three classes of classification, the definition of pending, is itself somewhat ambiguous, and so creates a huge amount of confusion in the results. For the feedback segment, the accuracy of the two-class classification was raised from 52.1% to 74.1%, which was not as significant as the executive segment, but nonetheless constitutes a large improvement. The classification result of the three-class classification is raised from 36.7% to 62.1%, which is a big improvement in comparison. We speculate that this is because the third class is overtime and has unique feedback compared to the other two classes. Overall, all classifiers, except for the three-class classification in the executive segment, show significant improvement in classification after our feature filtering method. This shows the effectiveness of our feature extraction method and objectively proves that our experimental paradigm activates different waveforms in different situations.

## 4. Conclusions and Discussion

In this study, we hypothesized that the EEG activity of the brain differs when a person performs correct or erroneous actions, or when he receives different feedback. We designed an experiment that activates ErrP and recruited 26 subjects. They participated in the experiment in an informed manner and started the formal experiment after 10 training trials. We collected his EEG data throughout the experiment.

We pre-processed the data for each individual. Specifically, we select channels to remove bad sectors, re-reference to remove the influence of reference electrode position, filter signals to remove power frequency infection, and select useful frequency bands. We then segment the data to extract the ERP, downsampling to subtract the amount of data, perform independent component analysis to remove artifacts and subtract baseline to mitigate the effects of regular brain activity. The signals before and after processing are shown in Figure 5 and Figure 7. We could assume our pre-processing to get excellent results.

We found that the data of different executions and feedbacks after pre-processing are already statistically different. This difference is present in the EEG data from our unique cerebellar regions (Figure 9). This result demonstrates that activated ErrP can be detected in the epidermis of cerebellar brain regions, and also suggests that the cerebellum is involved in cognitive activity to some extent. The difference in the execution segment is more significant than that in the feedback segment. This may be because subjects in the executive segment were more engaged, and their brain activity was more active. We found that the data varied considerably from subject to subject, with some subjects having richer variability in different situations than others, which we attribute to the different levels of concentration on the experiment and sensitivity to error of different subjects.

In the current study, we extract 15 standard time-frequency domain features for each channel in each trial for each subject. We select the channels that differ more in different cases using a cross-correlation approach and combine them with our extracted features to form a new feature matrix. We perform SBS on the new feature matrix, perform KNN classification several times in each iteration, remove the features that are least positive for classification, and finally extract the number of features and specific feature types that are most appropriate for classification. The final extracted best feature matrix was different for each subject, but all proved to be helpful for classification accuracy.

When classifying the data of the execution segment, we divided the situation into two or three classes depending on whether the pending status, meaning the key pressed before the appearance of the “STOP” sign, is considered. For the feedback segment, we divided the situation into two or three classes, depending on whether or not the independent case of overtime was included, meaning that the target appears but no key is pressed followed by the “overtime” feedback image. The final results are shown in Table 3. We can find good results for the two-class classification of the execution part. We could consider that at this point we are able to distinguish, with a high accuracy, whether the test set of the execution segment is correct or error. That is, when subjects pressed a key, there was a certain difference in thinking activity in the following hundreds of milliseconds when they were able to realize whether the operation was correct or an error. The three-class classification results are inferior and have no practical value. We believe that this is because in the state of pending, subjects think that they will think they are correct or error, rather than some independent mental model. For the feedback segment, the results of the two-class and three-class classifications are only passable, for our paradigm, the EEG data of subjects in different situations in this segment were somewhat differentiated but not significantly enough We speculate that because many trials of experiments do not require subjects to respond, subjects had low concentration and low thought activity during these segments, resulting in lower variability differences in these EEG data. But in either case, the accuracy achieved with the classifier using the screened features as input is better than that with the full features as input has been improved. This illustrates the usefulness of our feature extraction method. We also tried to concatenate the data from all subjects for an overall analysis. However, due to individual differences, the amount of conductive gel, etc., each subject’s data varies greatly, resulting in poor results for extracting features and classification of overall.

To summarize, our results support our hypothesis. Our main results are the discovery of ErrP in cerebellar regions similar to those in brain regions, as well as the proposal of a feature extraction method that is applicable to ERP with multiple conditions. To be more specific, our conclusions and contributions are listed as follows. For the experiments we designed, ErrP appear in each area of the brain and cerebellum when the subject performs an error action or sees an error feedback. For our experimental equipment and experimental paradigm, we propose a pre-processing process that can effectively eliminate various disturbances in EEG and obtain valid data, and this method may be generalized to other paradigms. Our proposed feature extraction method can improve the performance of the classifier. The two-class classification effect for the execution segment is the best, and the average accuracy of 80.5%, which has further research and practical value, such as integrated, instantaneous, BCI applications with self-help error correction mechanisms. The performance of the other classifiers is not satisfactory, but it is also better than when we use other classification methods directly without feature screening. Many topics remain open and could constitute further research: (1) The ability of data processing to effectively extract common features, making it possible to classify the data taken to each individual directly without the need to train each subject individually; (2) The classifier in the result segment has a low correct rate. Whether and how we can further improve it to enable the formation of BCI applications where the program performs different actions when subjects see different feedback pictures with little variance; (3) For the EEG caps and experiments we designed, the current data processing and classifier have good results, but its robustness and transferability need to be further demonstrated.

## Figures and Tables

**Figure 1 brainsci-12-01173-f001:**
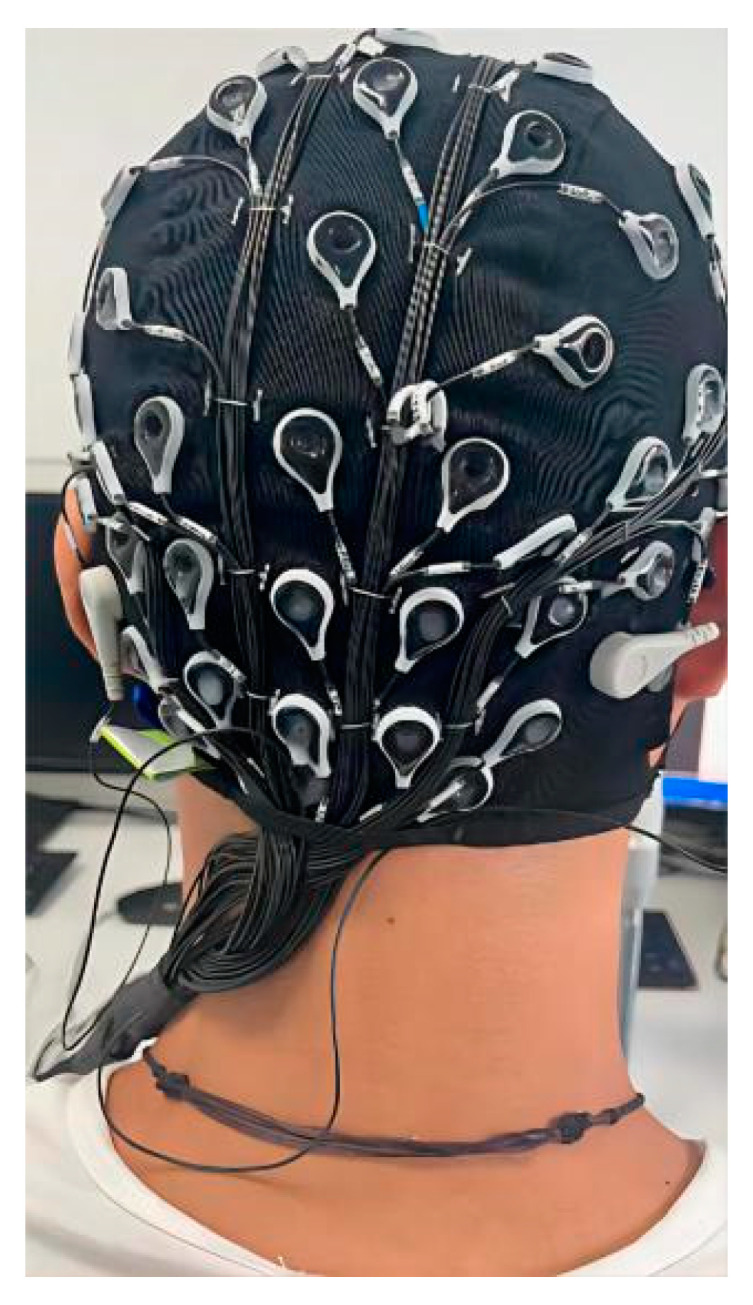
EEG cap containing cerebellar regions.

**Figure 2 brainsci-12-01173-f002:**
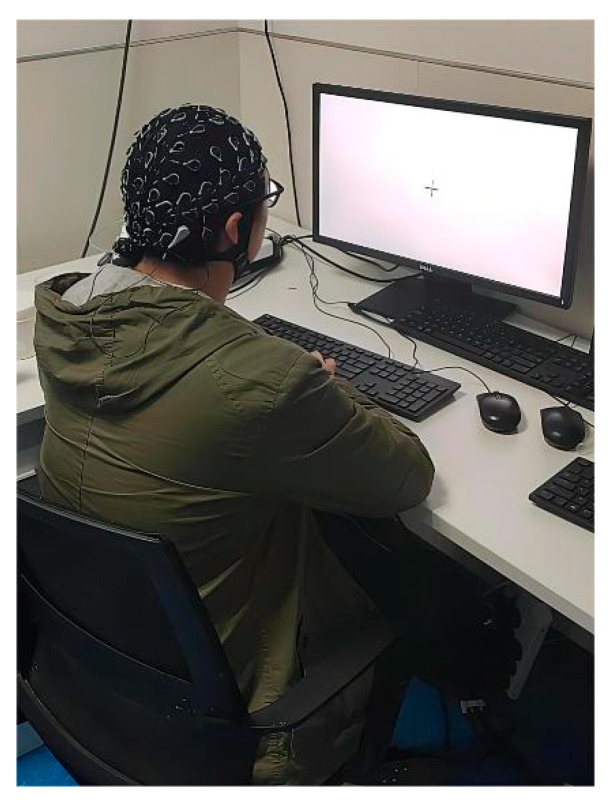
One of subjects is doing the experiment.

**Figure 3 brainsci-12-01173-f003:**
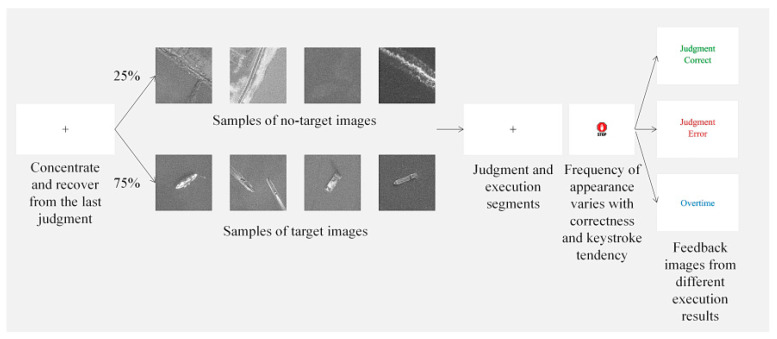
The process of each trial of experiments.

**Figure 4 brainsci-12-01173-f004:**
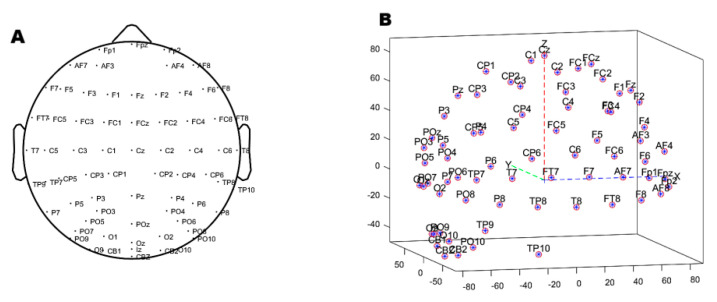
Electrode position. (**A**) 2D top view (**B**) 3D side view.

**Figure 5 brainsci-12-01173-f005:**
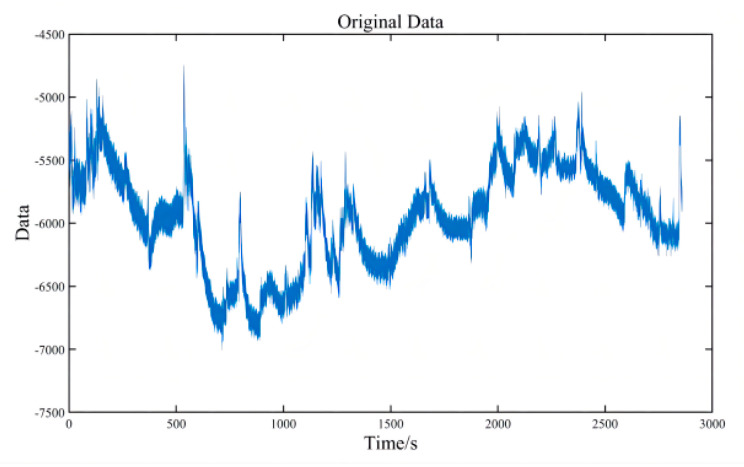
Raw data of CBz of the subjcet03.

**Figure 6 brainsci-12-01173-f006:**
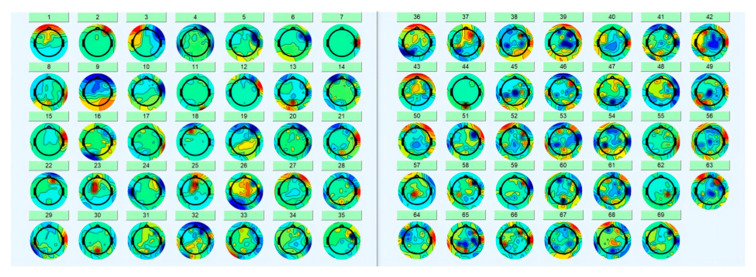
Topographic map of subject03 after independent component analysis.

**Figure 7 brainsci-12-01173-f007:**
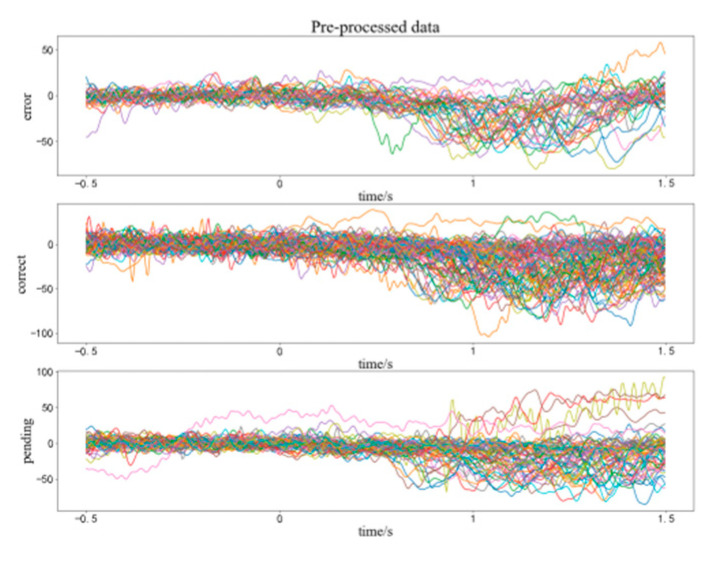
Processed execution waveform segment data of CBz of the subjcet03.

**Figure 8 brainsci-12-01173-f008:**
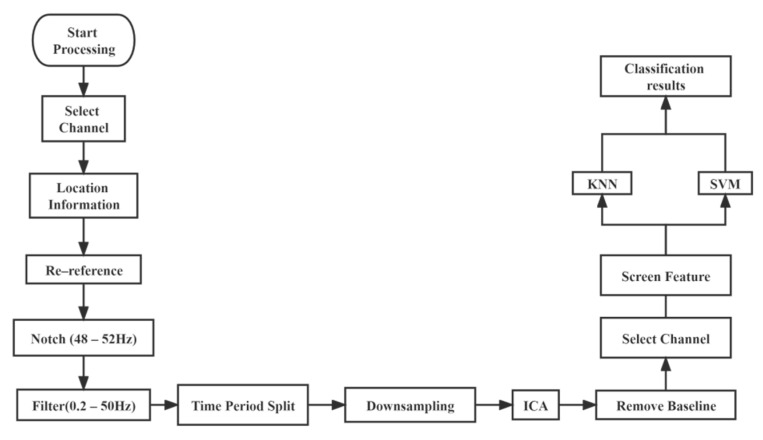
The flowchart for pre-processing, feature extraction and classification.

**Figure 9 brainsci-12-01173-f009:**
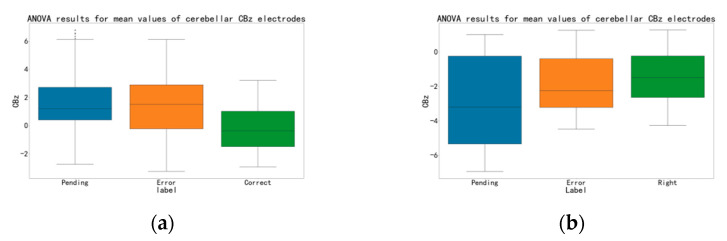
(**a**) The ANOVA result of each condition of the subject 03 electrode C4. (**b**) The ANOVA result of each condition of the subject 03 cerebellar electrode CBz.

**Figure 10 brainsci-12-01173-f010:**
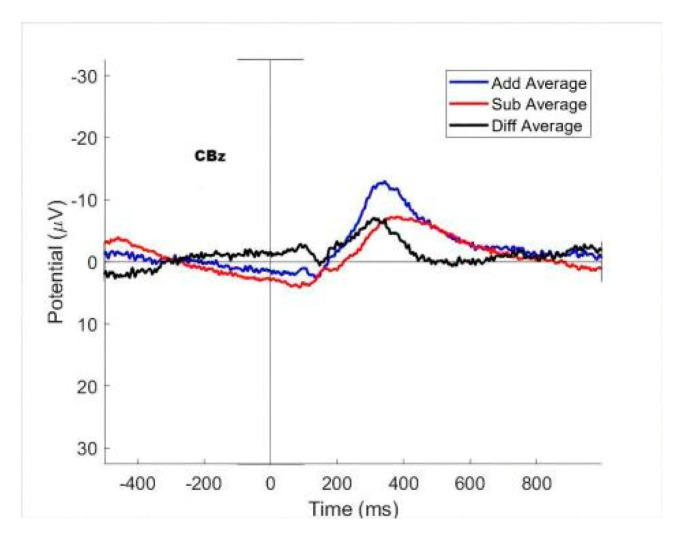
Cerebellum region CBz waveform comparison of multiple subjects.

**Figure 11 brainsci-12-01173-f011:**
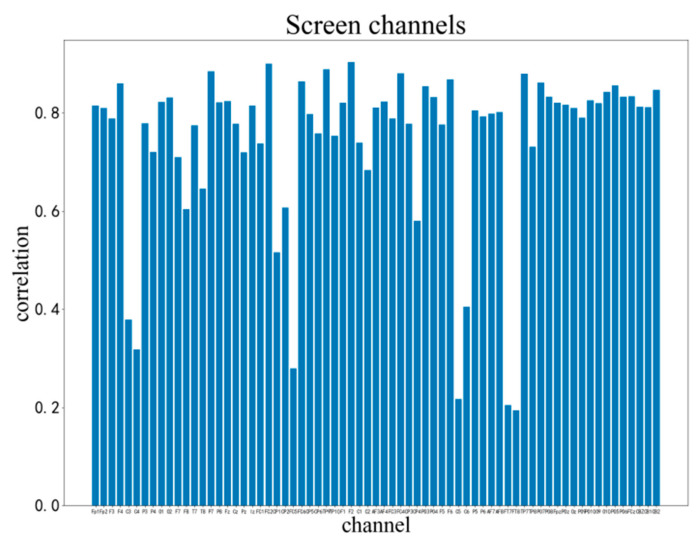
Correlation index of correct and error execution of each electrode of the subject 03.

**Figure 12 brainsci-12-01173-f012:**
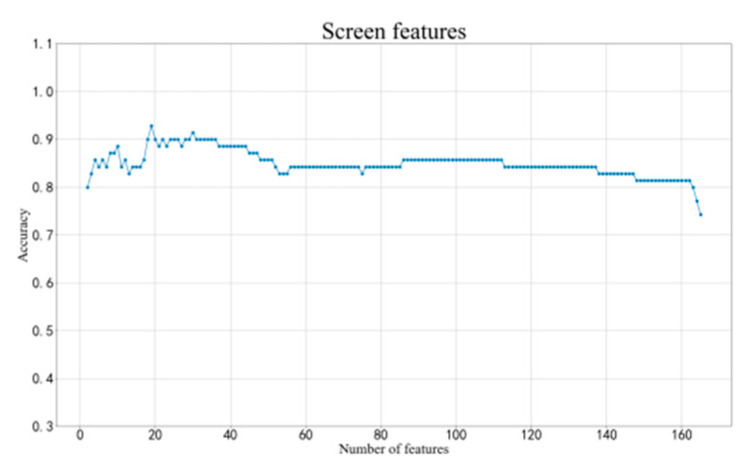
Three-class SBS classification curve of subject 03.

**Table 1 brainsci-12-01173-t001:** The average accuracy and standard deviation of the test set for multiple verifications of the subject 03 execution segment.

	SVM Pre-Screening	SVM Post-Screening	KNN Pre-Screening	KNN Post-Screening
Two-class Average Accuracy	45.2%	82.1%	34.1%	84.7%
Two-class Standard Deviation	0.104	0.062	0.081	0.048
Three-class Average Accuracy	34.1%	37.8%	33.2%	31.7%
Three-class Standard Deviation	0.083	0.053	0.069	0.046

**Table 2 brainsci-12-01173-t002:** The average accuracy and standard deviation of the test set for multiple verifications of the subject 03 feedback segment.

	SVM Pre-Screening	SVM Post-Screening	KNN Pre-Screening	KNN Post-Screening
Two-class Average Accuracy	51.5%	73.4%	63.9%	72.0%
Two-class Standard Deviation	0.019	0.025	0.033	0.028
Three-class Average Accuracy	47.2%	63.2%	57.9%	60.3%
Three-class Standard Deviation	0.042	0.009	0.036	0.024

**Table 3 brainsci-12-01173-t003:** Classification accuracy of execution and feedback of all subjects.

Subject	Execution				Feedback			
	Pre-Screening		Post-Screening		Pre-Screening		Post-Screening	
	Two-Class	Three-Class	Two-Class	Three-Class	Two-Class	Three-Class	Two-Class	Three-Class
01	55.1%	36.6%	73.4%	38.5%	48.2%	43.9%	64.1%	53.1%
02	52.6%	40.4%	81.5%	41.1%	60.3%	45.8%	77.3%	48.0%
03	45.2%	34.1%	82.1%	37.8%	51.5%	47.2%	73.4%	63.2%
04	36.7%	30.0%	68.8%	33.3%	52.6%	38.9%	71.1%	55.9%
05	59.3%	40.5%	85.3%	55.2%	53.8%	29.0%	71.7%	56.1%
06	49.2%	38.1%	77.3%	39.6%	51.4%	41.3%	72.3%	62.9%
07	58.8%	37.7%	86.7%	26.8%	45.9%	45.1%	83.4%	61.3%
08	49.6%	35.4%	74.0%	43.5%	53.6%	35.7%	78.6%	59.2%
09	55.3%	41.2%	85.4%	42.4%	46.4%	37.4%	68.2%	63.1%
10	50.0%	37.9%	84.8%	36.7%	51.4%	34.5%	77.0%	62.6%
11	38.8%	25.4%	79.6%	43.7%	41.6%	40.4%	84.4%	47.1%
12	42.9%	35.8%	83.6%	35.7%	53.2%	30.2%	72.8%	64.2%
13	46.7%	28.4%	85.1%	42.1%	43.1%	26.7%	78.2%	48.2%
14	53.5%	36.4%	71.7%	41.3%	49.3%	30.0%	73.7%	47.3%
15	51.7%	34.7%	86.0%	27.4%	50.4%	31.4%	82.3%	52.9%
16	46.8%	41.2%	82.5%	40.1%	47.5%	35.0%	71.2%	62.5%
17	56.2%	35.4%	77.4%	51.3%	60.4%	28.9%	75.0%	51.3%
18	48.1%	36.5%	90.2%	25.5%	51.2%	37.1%	86.5%	61.7%
19	44.2%	38.9%	80.0%	42.0%	49.2%	33.7%	67.2%	58.2%
20	61.0%	39.6%	68.9%	45.2%	57.4%	41.4%	58.7%	53.6%
21	38.4%	34.5%	78.3%	34.6%	48.6%	32.5%	70.1%	58.1%
22	57.1%	42.6%	81.6%	41.2%	54.2%	37.9%	66.1%	57.4%
23	42.4%	36.1%	85.1%	42.6%	59.4%	39.8%	71.2%	64.2%
24	47.2%	33.8%	79.2%	35.7%	52.4%	36.4%	74.1%	63.6%
25	60.0%	38.0%	84.7%	43.3%	61.3%	45.2%	78.4%	60.3%
26	56.7%	27.6%	78.3%	48.2%	59.3%	37.9%	71.2%	59.5%
average	50.1%	36.0%	80.5%	39.6%	52.1%	36.7%	74.1%	62.1%

## Data Availability

The data are available upon reasonable request.

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
