# Peer review of "A Method for the Study of Cerebellar Cognitive Function—Re-Cognition and Validation of Error-Related Potentials"

_brainsci, 2022, doi:10.3390/brainsci12091173_

Round 1

Reviewer 1 Report

I have reviewed the manuscript titled “A method for the study of cerebellar cognitive function — re- 2 cognition and validation of error-related potential”. The overall contents of this manuscript are not well organized to give a clear overview of this work. I have suggested some comments about this work are as the following:

Comments to the Authors:

1.     Authors should write clearly abstract clearly including, background, method, results, significance.

2.     All the figures resolution is very weak. It should be updated with high resolution at least 300db.

3.     In method section author should add one more figure related to the system architecture diagram/flowchart for EEG signal process.

4.      My suggestion is that the authors should write discussion section clearly in more details like how and why this study is important than previous clinical research, like “Chikara RK, Chang EC, Lu Y-C, Lin D-S, Lin C-T and Ko L-W (2018) Monetary Reward and Punishment to Response Inhibition Modulate Activation and Synchronization Within the Inhibitory Brain Network. Front. Hum. Neurosci. 12:27. doi: 10.3389/fnhum.2018.00027”.

5.     Authors should write clearly conclusion of this study.

6.     The authors should write some limitations of this study and application in more details.

Reviewer 2 Report

In order to determine if the cerebellum is engaged in cognition, the authors of the paper “A method for the study of cerebellar cognitive function — recognition and validation of error-related potentials” conduct an EEG analysis. Although the topic is incredibly intriguing, the paper's text, structure and scientific sound is awkward. I'll give you a lot of feedback.

1.     In line 12 “ErrP” has not been specified jet

2.     I would suggesting refocusing and better addressing abstract, (e.g boosting information and following a chronological line in provide messages).

3.    Lines 47-48 The point emphasized is significant, although the language may be better handled.

4.     Line 66 makes hazardous assumptions

5.  I'm interested in the distinction between Figures 1 and 2's scientific differences. I suggest taking out Figure 2.

6.     In line 70 “ct” is a typos

7.     In line 74 “We designed, produced and used the EEG cap as” You may create a brand-new configuration because you are utilizing a commercial EEG headset. Please adjust the focus of this statement.

8.     In line 79 “His EEG”

9.     In line 80 “Each person performed 500 80 experiments of easily induced ErrP as in Figure 2.”.  It may be a task and not experiment?

10.  I would advise redesigning figure 3, as it contains little scientific information and is of low resolution. Leverage from the literature's description of the BCI protocol design timeframe.

11.  I would advise properly addressing the topic in line 94.

12.  Lines 114–119 might use some improvement and refocusing because they are difficult to understand.

13.  I would suggest pruning “handicap were reported”

14.  How do the authors define "psychological conditions" in line 126?

15.  I would suggest pruning “We call them subjects 01 to 26.”

16.  It is important to adequately address and contextualize the meaning of lines 132 to 134.

17.  Saying "SP1 and SP2 were free electron-134 trodes, which were not included in our data analysis" is better avoided. Thus is better to say: we used 69+2 channels.

18.  What information does figure 4b in this study provide? What does it signify scientifically? Figure with low resolution and weak caption

19.  Lines 139 and 140 should be combined and refocused, in my opinion.

20.  Why does "Read by the extension function of the EEGLAB 14.0 toolbox of MatLab R2020b." signify anything?

21.  Figure 5's structure is awkward. I can't seem to figure out what it means.

22.  Lines 176–178 include hints that are backed by literary works. Please cite this.

23. I would propose describing classifiers, parameters, and hyperparameters in lines 192-194. I'd advise using this document to your advantage “A Machine Learning Approach Involving Functional Connectivity Features to Classify Rest-EEG Psychogenic Non-Epileptic Seizures from Healthy Controls”

24.  In line 198 what X-axis, Peak to Peak Distance, and zero-crossing rate are ? How was the calculation performed?

25.  How is the assertion in lines 204–205 inferred?

26.  I would suggest to better address lines 221-224

27.  Eq 1 is weird since y is binary although there are three classes.

28.  You are doing a multiclass analysis subject-based classification. To determine the result, I suggest utilizing ANOVA or bootstrap analysis.

29.  Figures 8, 10, 11, 12, and 13 should be pruned since they are awkward and make no sense. ANOVA or bootstrap analysis with boxplots and one nice table could be a solution to grasp what you are testing.  I'd advise using this document to your advantage “A Machine Learning Approach Involving Functional Connectivity Features to Classify Rest-EEG Psychogenic Non-Epileptic Seizures from Healthy Controls”

30.  Are the x-axes in Figure 10 a connected to the EEG channels?

31.  I would advise rewriting the results section.

The conclusion and findings need to be better handled, but it has to be rewritten after applying the suggestions above

Round 2

Reviewer 2 Report

This paper was greatly improved by the authors. 

I recommend accepting after two more comments. 

To improve the quality of the figures, I suggest using a "FontSize," at list 14, and a "fontweight," "bold" for labels, legends, and axes, etc.

Additionally, I suggest using the EEGLAB function "topoplot(datavector,'eloc file')" + "colorbar" or a plot similar to this to illustrate Figure 11.